# DISCRETIZATION OF CONTINUOUS INPUT SPACES IN THE HIPPOCAMPAL AUTOENCODER

## ABSTRACT

Understanding the encoding mechanisms of hippocampal place cells remains a significant challenge in neuroscience. Although sparse autoencoders have been shown to exhibit place cell-like activity, the underlying processes are not fully understood. In this study, we compare spatial representations learned by dense and sparse autoencoders trained on images of 3D environments and find that only sparse autoencoders with orthonormal activity regularization in latent space produce place cells. We then show that this regularization promotes similar images to map onto the same neurons, acting as a locality-sensitive hash function. Notably, we demonstrate that these neurons are visually interpretable through activity clamping and decoding, suggesting the formation of detailed episodic memories at the single-neuron level. We then introduce a novel metric to quantify how neurons discretize the image space into disjoint receptive fields, revealing that sparse autoencoders tile input spaces with minimal overlap. Furthermore, we observe that whereas dense autoencoders generate population codes resembling visual cortex activity near criticality, sparse autoencoders produce higher-dimensional codes, thus suggesting a similar coding strategy in the hippocampus. Extending our approach to the auditory domain, we also replicate the emergence of "frequency place cells" by training sparse autoencoders on audio snippets sampled from a frequency-varying signal, and show that population representations retain the statistical structure of the sample distribution. Lastly, we demonstrate that reinforcement learning agents can leverage these high-dimensional image representations to solve complex spatial-cognitive tasks, despite their inherent brittleness. Overall, our findings elucidate how sparse input compression in autoencoders can give rise to discrete, interpretable memories, establishing an explicit link between episodic memory formation and spatial representations in the hippocampus.

## 1 INTRODUCTION

Early physiological experiments with rats revealed that certain neurons in the hippocampus exhibit increased activity when the animal occupies specific regions of the environment O'Keefe & Dostrovsky (1971). Ever since the discovery of such "place cells", decades of animal research have established the hippocampus as a neural system that learns a cognitive map of the environment and uses it for spatial navigation (Moser et al., 2008). Subsequent experimental studies also identified the hippocampus as a key structure in episodic memory formation (Moser et al., 2015). Although several attempts have been made to unify these observations under a coherent conceptual framework (Redish, 1999; Eichenbaum, 2017), a clear mechanistically relationship episodic memory and spatial representations remains elusive. Furthermore, numerous experiments reporting the instability of place cell activity over time and their modulation by other non-spatial variables (Fenton & Muller, 1998; Jercog et al., 2019) raise an open question: what are these cells truly encoding?

Efforts to answer this question have demonstrated that place cell-like activity can emerge under various conditions: when artificial agents optimize a predictive coding objective (Recanatesi et al., 2021; Uria et al., 2020; Ratzon et al., 2023; Gornet & Thomson, 2023; Levenstein et al., 2024; Chen et al., 2022), when networks optimize temporal stability and pairwise decorrelation in processing visual inputs (Wyss et al., 2006), or when building sparse, compressed representations of environmental states (Santos-Pata et al., 2021a;b; Benna & Fusi, 2021; Ketz et al., 2013). Notably, the approach where sparse compression of information leads to spatial tuning aligns with the earlier hippocampal

autoencoder model (Gluck & Myers, 1993), and has been shown to replicate several distinct place cell phenomena following environmental manipulations (Santos-Pata et al., 2021a;b).

In this work, we further investigate the mechanisms behind episodic memory formation and the emergence of place cells in the hippocampal autoencoder model. We demonstrate that sparse autoencoders equipped with orthonormal activity regularization can create discontinuities in the manifold of the latent space, discretizing arbitrary input spaces into disjoint receptive fields, whereby subsets of similar inputs converge onto distinct neurons. When applied to visual images, this clustering process generates place cells operating on a very high-dimensional population code. In turn, these neurons are shown to encode detailed visual memories. Moreover, we show that similar effects result from applying the same principle in the auditory domain, recapitulating recently reported "frequency place cells". Lastly, we show that reinforcement learning agents can make use of such sparse and high-dimensional hippocampal-like representations to solve spatial-cognitive tasks.

## 2 MODEL AND RESULTS

### 2.1 HIPPOCAMPAL-LIKE PLACE CELLS EMERGE IN SPARSE AUTOENCODERS

We studied the learning of spatial representations by training autoencoders (Figure 1a) with randomly sampled images from four different tasks in the Animal-AI environment (Beyret et al., 2019): Double T-maze, Cylinder, Object Permanence, and Thorndike. We trained two types of autoencoders. "Dense" autoencoders aimed solely to reconstruct the input images, thus preserving input information in their latent space $Z$. In contrast, "sparse" autoencoders had an additional objective beyond input reconstruction: to develop sparse activity patterns in the latent space $Z$. This was achieved using the following loss function:

$$\mathcal{L} = \frac{1}{m}\|\mathbf{X} - \hat{\mathbf{X}}\|_2^2 + \frac{\lambda}{mn}\|\mathbf{I}_n - \mathbf{Z}^\mathrm{T}\mathbf{Z}\|_\mathrm{F}, \tag{1}$$

where $m$ denotes the batch size, $n$ the number of neurons in $Z$, and the first term is the mean squared error (MSE) between inputs $\mathbf{X}$ and their reconstructions $\hat{\mathbf{X}}$, encouraging $Z$ to preserve input information. The second term is an orthonormal activity regularization term, whose strength is controlled by $\lambda$, pushing the Gramian $\mathbf{Z}^\mathrm{T}\mathbf{Z}$ towards the identity matrix $\mathbf{I}_n$. Since $\mathbf{Z}^\mathrm{T}\mathbf{Z}$ captures the co-activation strengths between neurons in a training sample batch, the orthonormal activity regularization promotes pairwise decorrelation while ensuring equal contribution across neurons. We found this approach yields improved and more reliable results compared to the L1 activity regularization term typically used in sparse autoencoders, particularly in alleviating the dead ReLU problem (Lu et al., 2019). For dense autoencoders, $\lambda$ was set to zero, leaving only the reconstruction error. We refer the reader to the Detailed methods section in the Appendix for a complete description of the environments, dataset generation, and parameters used in this study.

Training both types of autoencoders yielded significantly different internal representations of space in their latent space. Dense autoencoders developed many neurons that fired almost everywhere in space, with no defined place fields. In contrast, sparse autoencoders developed a majority of neurons with one or two localized place fields, similar to place cells in the hippocampus (Figure 1b, c). The spatial specificity of sparse autoencoder neurons was also reflected in significantly higher spatial information scores compared to dense autoencoder neurons (Figure 1d). These results demonstrate that single-unit spatial tuning emerges in sparse autoencoders but not in dense autoencoders, despite both types of networks containing the same amount of positional information at the population level, as shown by linear decoding analyses (Figure 1e).

### 2.2 SPARSE AUTOENCODERS DISCRETIZE AND TILE THE IMAGE SPACE WITH INTERPRETABLE NEURONS

Identifying neurons with spatial selectivity similar to hippocampal place cells allowed us to investigate what these neurons encode. Given that their spatial selectivity must arise from some form of visual selectivity, we explored whether they also exhibit localized receptive fields in image space.

We created an image space by extracting semantically-relevant image embeddings of all samples using CLIP and further reducing the dimensionality to a 2D space with UMAP. We then searched for clusters in this 2D image space by running the DBSCAN algorithm on the points corresponding

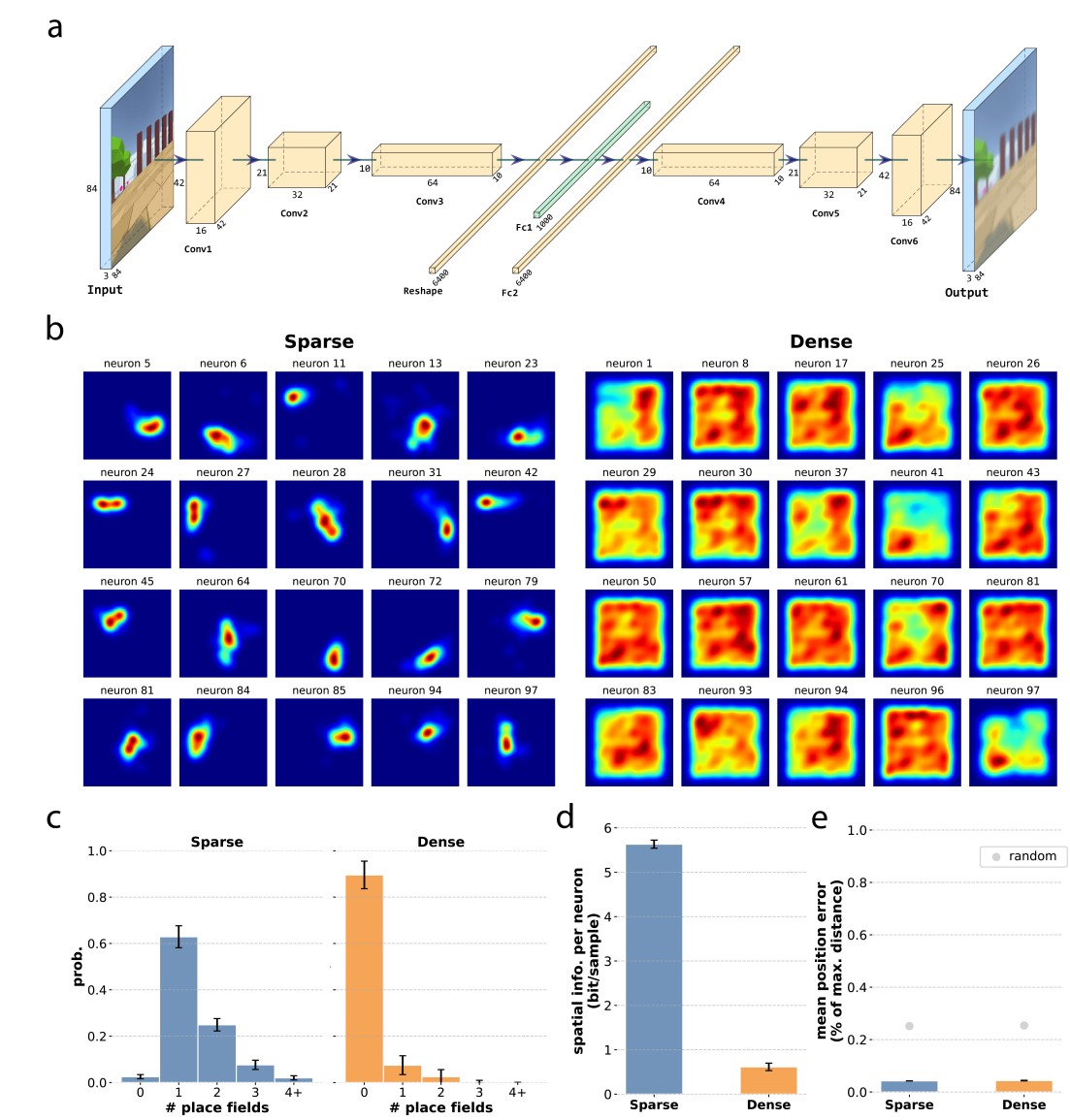

Figure 1: Hippocampal-like place cells emerge in sparse autoencoders. (a) Autoencoder architecture, featuring the hidden layer or latent space $Z$ (denoted as Fc1) with 1000 neurons. (b) Representative examples of the neurons' spatial ratemaps for sparse and dense autoencoders. (c) Probability distribution of place field number across environments. (d) Average spatial information per neuron across environments. (e) Normalized average distance error of linear decoding of position with the ratemaps' population vectors, across environments. The grey dots represent the expected linear decoding errors after performing 1000 random permutations of the ratemaps' values.

to images that maximally activated a particular neuron (see Figure 2a and Detailed methods in the Appendix). These clusters formed convex hulls (i.e., patches) that corresponded to the neuron's receptive fields in the image space. When pooled together, receptive fields across neurons partitioned and covered the entire image space (see example in Figure 2b). Furthermore, we computed an overlap metric to estimate the redundancy across the neurons' receptive fields. We observed that sparse autoencoder neurons tiled the image space in a minimally-overlapping manner, in contrast to dense autoencoder neurons, whose overlap tended to be significantly higher (Figure 2c).

Additionally, we performed unit clamping experiments, setting neurons to their maximal recorded value while others were set to zero, and then decoded their activity back to images. The generated images showed a striking resemblance to the training images, making these neurons highly interpretable (Figure 2d). These results establish a solid relationship between episodic memory formation and spatial coding.

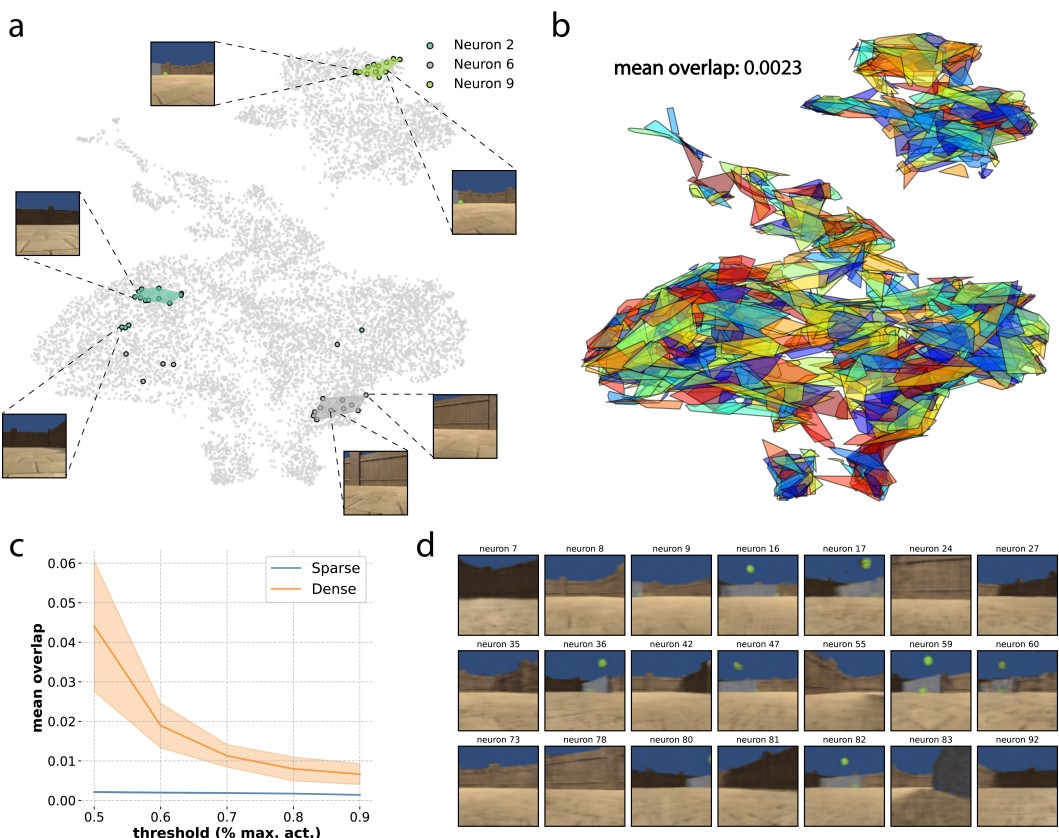

Figure 2: Sparse autoencoders discretize and tile the image space with interpretable neurons. (a) Images taken in one of the environments ('Cylinder'), encoded with CLIP and further reduced to two dimensions with UMAP. Points of different colors correspond to the images that maximally activate each example neuron (above the 50% threshold of the maximum neuron's recorded activity). Clusters of maximally activated images are extracted with DBSCAN, making up the convex hulls. (b) Convex hulls for all neurons in a sparse autoencoder trained with images from the 'Cylinder' environment. The overlap metric corresponds to the expected overlapping area (in %) of two randomly chosen hulls (see Detailed methods in the Appendix for further details). (c) Average overlap in 2D image space of sparse and dense autoencoders, across tasks and for a range of threshold values of maximal activation. (d) Example interpretable neurons in the sparse autoencoder. The corresponding neuron in latent space $Z$ is set to its maximum recorded value across the dataset, while all other neurons are set to zero. Then, the enforced activity vector $Z$ is deconvolved into an image by passing it through the decoder.

## 2.3 HIGH-DIMENSIONAL POPULATION STRUCTURE IN SPARSE AUTOENCODERS

Having linked the formation of episodic memories with the discretization of the image space in sparse autoencoders, we explored the population structure of the latent space representations. Inspired by Stringer et al. (2019) on the dimensionality of the population code in the mouse visual cortex, we examined the dimensionality in our autoencoders. Dimensionality was estimated by performing PCA on $Z$ and computing the linear fit of the resulting eigenspectrum in log-log space, yielding a power-law exponent, $\alpha$. High $\alpha$ values indicate low-dimensional codes, while low $\alpha$ values suggest

high-dimensional codes. An $\alpha \approx 1$ indicates a criticality regime with a high-dimensional but smooth (i.e., no discontinuities) underlying manifold, as seen in neural responses in the visual cortex (Stringer et al., 2019).

We found that dense autoencoders had dimensionality scores close to 1, similar to visual cortex (Stringer et al., 2019), whereas sparse autoencoders exhibited higher-dimensional representations (Figure 3a), aligning more with the efficient coding hypothesis (Barlow et al., 1961). The almost-flat eigenspectrum suggests that sparse autoencoders' population activity indeed encodes fine stimulus features. Moreover, the orthonormal activity regularization also disrupted the input-output similarity preservation typically seen in dense autoencoders (Figure 3b), further supporting the idea of a sharp discretization of the image space by sparse autoencoders.

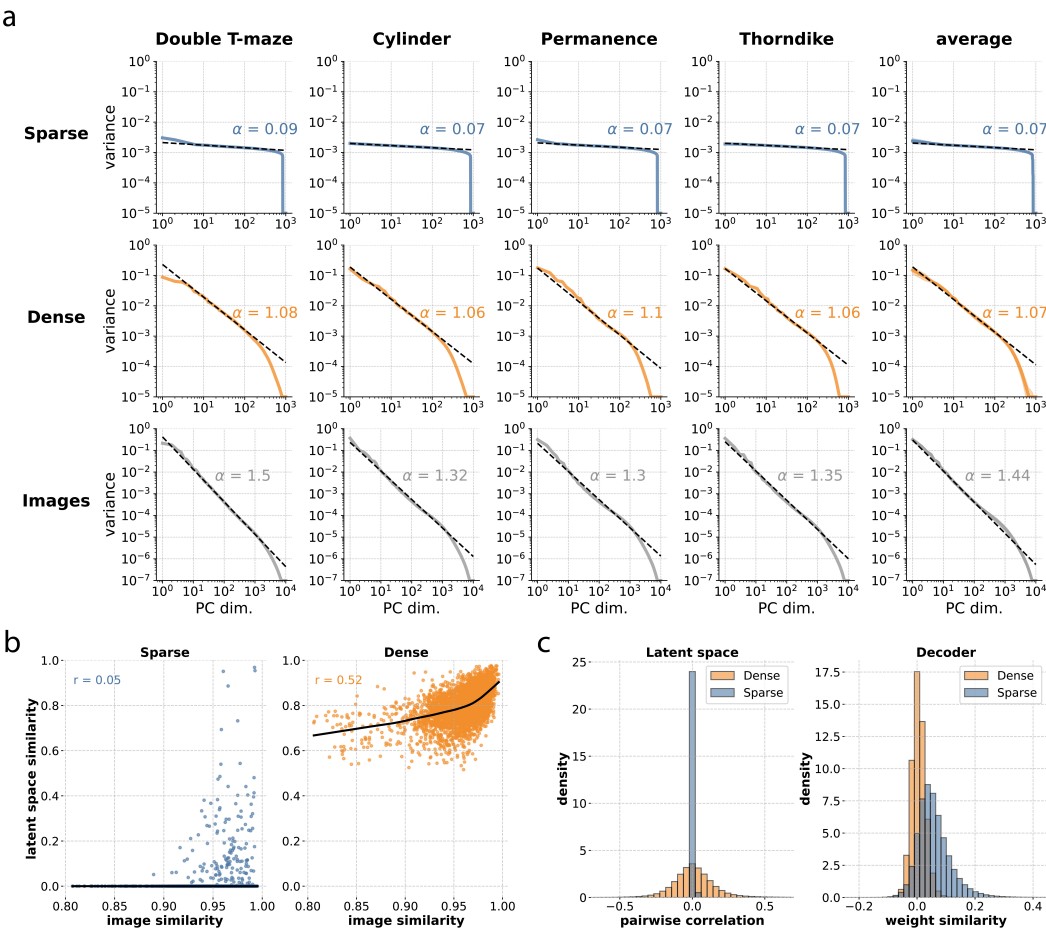

Figure 3: High-dimensional population structure in sparse autoencoders is grounded on mixed selectivity. (a) Eigenspectrum decay in latent space representations (first two rows) and images from the environments (third row). Parameter $\alpha$ corresponds to the power law exponent from linear fitting in log-log space. (b) Input-output similarity for sparse and dense autoencoders, with data pooled across environments. Correlation scores correspond to Spearman's rank coefficients, and fitting curves have been generated with a locally weighted scatter-plot smoother (LOWESS) for improved visualization. (c) Pairwise Pearson correlation scores between all neurons' activity in latent space, pooled across environments (left) and pairwise kernel similarity in the decoder weights (layer Fc2), representing the similarity density across "words" in the learned "dictionary" (right).

Borrowing concepts from sparse dictionary learning (Lewicki & Sejnowski, 2000), we considered the decoder weights to be the dictionary of kernels, and the sparse neuron activities to be the coefficients

that use the dictionary to reconstruct the inputs. Dense autoencoders exhibited orthogonal kernels, while sparse autoencoders showed non-orthogonal kernels despite highly uncorrelated activity patterns (Figure 3c). This indicates that orthogonal activity does not imply orthogonal kernels, and that sparse autoencoder neurons learned similar feature combinations, indicative of mixed selectivity in neurons (Fusi et al., 2016). These findings suggest that mixed feature selectivity and high dimensionality are closely linked to the formation of detailed episodic memories.

### 2.4 ZERO-SHOT LEARNING OF PLACE CELLS IN SPARSE AUTOENCODERS

A typical observation in the hippocampus is that place cells can be identified within the first minutes of an animal being exposed to a new environment (Frank et al., 2004). Given that we have shown that sparse autoencoders exhibit very high dimensionality (Figure 3a) and single neurons tend to encode small clusters of samples (Figure 2), we investigated the extent to which neurons that learned to encode samples in one environment could generalize to encoding unseen environments and exhibit zero-shot place cells. Therefore, we trained sparse autoencoders in one environment and tested them across all others. Strikingly, neurons developed place fields with distributions very similar to those in their training environment (Figure 4a). Furthermore, the average spatial information across neurons and the mean decoding error of the rate maps were very similar, with no significant degradation compared to the training environment (Figure 4b). These results suggest that the network's circuitry learned to cluster similar samples onto single neurons in a more generic manner, beyond the specific details of the training data.

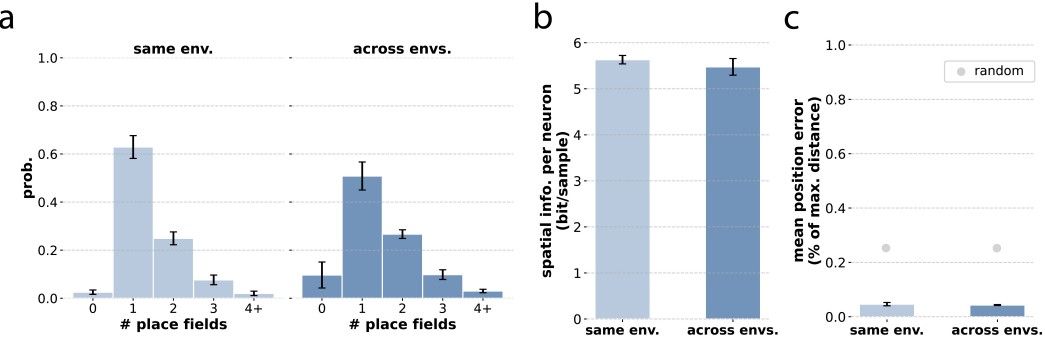

Figure 4: Zero-shot place cells in sparse autoencoders. (a) Probability distributions of place field number when testing a model within its training environment (light blue) or across unseen environments (dark blue). (b) Average spatial information per neuron, pooled across models and testing environments. (c) Normalized average distance error of linear decoding of position with the ratemaps' population vectors, across models and testing environments. The grey dots represent the expected linear decoding errors after performing 1000 random permutations of the ratemaps' values.

### 2.5 SPARSE AUTOENCODERS DISCRETIZE AND TILE THE INPUT FREQUENCY SPACE IN AN EXPERIENCE-DEPENDENT MANNER

If the hippocampus functions as a generic, modality-independent episodic memory system, our findings with the sparse autoencoder should generalize to other input modalities, such as sound. Indeed, "place cell"-like activity in the hippocampus has been reported for tasks involving "navigating" the sound frequency space, with neurons developing localized receptive fields around particular sound frequencies (Aronov et al., 2017). To investigate whether a similar effect could be observed within our framework, we trained autoencoders to compress and encode sound waves uniformly sampled from a linearly-varying frequency signal (Figure 5a).

We observed the emergence of frequency-specific receptive fields in sparse autoencoders, but not in dense autoencoders (Figure 5b), reproducing the main observations in Aronov et al. (2017). These receptive fields tiled the entire frequency space in a linear manner. However, when sampling was biased towards certain frequencies, the neurons' receptive fields became denser and clustered around those frequencies, preserving the statistical structure of the sample distribution (Figure 5d).

Additionally, similar to our previous findings with visually interpretable neurons, we found that individual neurons encoded particular frequencies so that these representations were readily decodable via activation clamping (Figure 5c). These results demonstrate that sparse autoencoders can discretize arbitrary input spaces to support episodic memory formation.

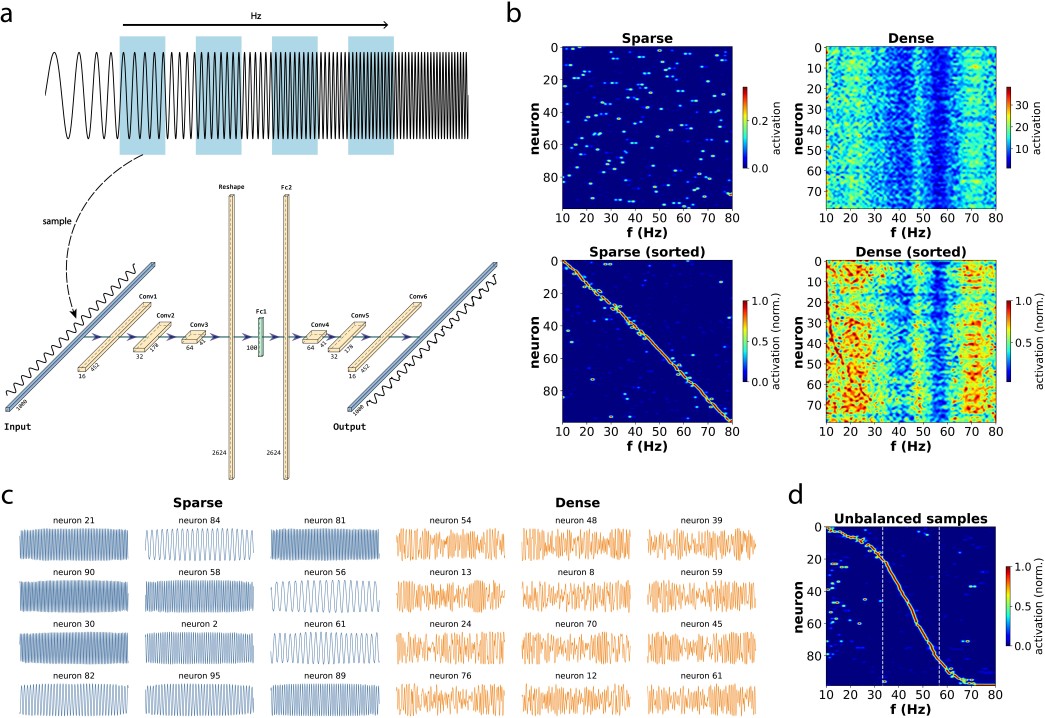

Figure 5: Sparse autoencoders discretize and tile the input frequency space in an experience-dependent manner. (a) Data samples are generated by applying a uniformly distributed sliding window to a linearly-varying frequency signal. The samples are fed into a convolutional autoencoder, analogous to the one used for vision (more details can be found in the Detailed Methods section of the Appendix). (b) Unsorted and sorted receptive fields by peak activity location for both sparse and dense autoencoders. Latent space activity $Z$ responding to pure tone test inputs was convolved with a Gaussian kernel (sigma of 0.5 Hz), and then normalized by the maximum per neuron in the sorted plots. Lanczos interpolation was applied to all plots for improved visualization. (c) Decoded output signals after setting the corresponding neuron in latent space to its maximum recorded value across the dataset, while all other neurons were set to zero. (d) Sorted receptive fields in a sparse autoencoder trained with an unbalanced dataset. The data samples were generated with a sliding window that was not uniformly distributed in the frequency space, but whose density followed a Gaussian distribution centered at 45 Hz. Dashed vertical lines denote one standard deviation $\sigma$ from the mean.

## 2.6 REINFORCEMENT LEARNING AGENTS LEARN EFFECTIVELY WITH SPARSE, HIGH-DIMENSIONAL REPRESENTATIONS

Representations used to build episodic memories are likely also employed for behavioral learning in the brain. Therefore, we investigated whether hippocampal-like representations emerging in sparse autoencoders would be suitable for reinforcement learning. To test this, we employed Deep Q-Networks (DQNs) (Mnih et al., 2015) incorporating either sparse or dense autoencoders to solve a range of tasks in the Animal-AI environment, which inherently require spatial navigation skills (see Figure 6a, and Detailed methods in the Appendix for further details on the tasks, model, and training parameters).

Very high-dimensional representations (such as those based on efficient coding) are known to be highly sensitive to slight input perturbations and are thought to generalize poorly to new, unseen

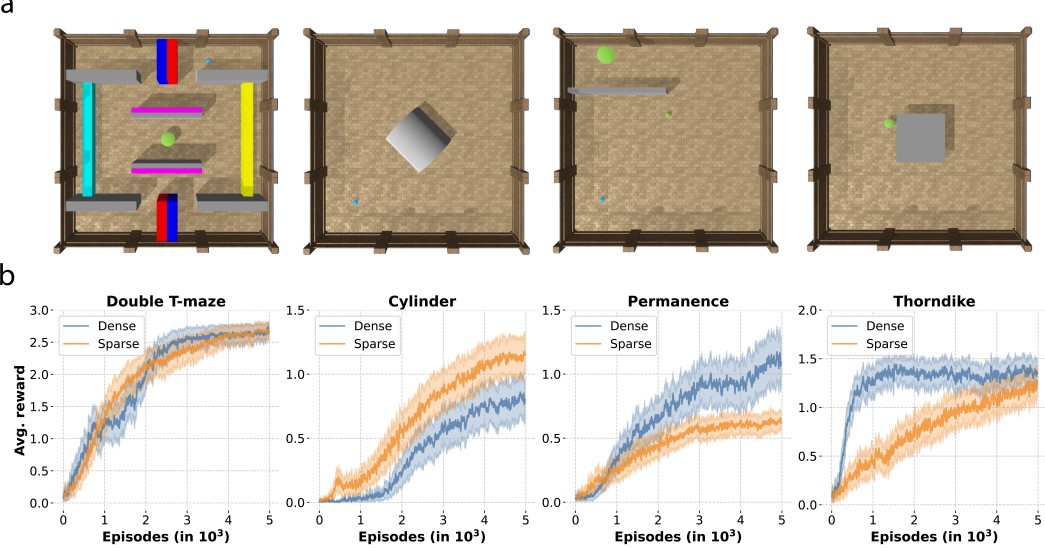

Figure 6: DQN agents learn effectively with sparse, very high-dimensional representations. (a) Overhead images captured from above the virtual arena of the four tasks (from left to right): Double T-maze, Cylinder, Object Permanence, and Thorndike. (b) Performance (average reward across episodes) of DQN agents using sparse and dense autoencoders across tasks.

samples (Nassar et al., 2020). Thus, one might expect that DQNs would struggle with tasks requiring generalization to unseen samples during training in non-stationary environments. Contrary to this expectation, we found that DQNs using sparse autoencoders were not systematically worse than those using dense autoencoders (Figure 6b). Although sparse autoencoders seemed to perform worse than dense autoencoders in two of the four tasks tested here (Object Permanence and Thorndike), they were superior in one of them (Cylinder) and had matched performance in the remaining one (Double T-maze). Therefore, while further testing is definitely needed to obtain a more reliable picture of their relative performance, these results suggest that, in practice, hippocampal-like representations can be suitable for reinforcement learning, despite their high dimensionality and inherent brittleness.

## 3 DISCUSSION

**Optimization objectives underlying place cell emergence** We have demonstrated the distinct emergence of place cells in autoencoders with orthonormal activity regularization (Figure 1). Notably, sparse compression alone was sufficient to develop spatial tuning, without the need for a predictive objective (Recanatesi et al., 2021; Ratzon et al., 2023; Uria et al., 2020; Gornet & Thomson, 2023; Levenstein et al., 2024; Chen et al., 2022). While predictive coding may explain other features of the hippocampus, such as place-cell theta sequences (Dragoi & Buzsáki, 2006), prediction does not appear to be necessary for the emergence of realistic place cells. We speculate that models optimized for next-input predictions likely learn compressed representations of the environment implicitly as part of the predictive objective. Furthermore, by training the autoencoders with randomly sampled and shuffled images, we have shown that neither temporal contiguity of samples nor any temporal correlations are required to develop place cells. This finding suggests that while predictive learning capturing temporal input correlations might correspond to experience-dependent theta sequences in the hippocampus, the formation of compressed state representations might correspond to time-independent learning processes at the gamma frequency scale (Lisman, 2005). Additionally, we have demonstrated that sparse autoencoders can learn localized receptive fields of the input space while breaking the relationship between input similarity and latent space similarity. This finding contrasts with previous research that emphasized the preservation of input-output similarity matching for learning localized receptive fields (Sengupta et al., 2018; Qin et al., 2023). Overall, it appears that

sparse compression alone is sufficient to learn localized receptive fields, that in turn manifest as place cells when applied to the visual domain.

**Sparse and very high-dimensional codes**   A slowly-decaying eigenspectrum, where fine details are over-weighted, represents codes that create discontinuities by disrupting the locality of the manifold structure supporting the input space distribution (Nassar et al., 2020). We demonstrated that this discontinuity can be induced by an orthonormal activity regularization objective, facilitating the creation of event memories by discretizing the image input space (Figure 2). Our results suggest that very high-dimensional codes underlie the formation of place cells (Figure 3), aligning with the efficient coding hypothesis (Barlow et al., 1961), which posits that the brain maximizes information by eliminating correlations in sensory inputs, leading to sparse coding (Olshausen & Field, 1996). Indeed, it has been shown that hippocampal neurons in rodents become sparser with prolonged exposure to the environment (Ratzon et al., 2023). Moreover, the storage of social memories in mice has been linked to high-dimensional representations in the hippocampus (Boyle et al., 2024). Importantly, sparsity has been shown to control a generalization-discrimination trade-off (Barak et al., 2013), which could explain why the hippocampus relies on sparse representations, well-suited for progressive discrimination of similar environments and events. Our study shows that smooth place cell maps can coexist with and emerge from extremely high-dimensional sparse codes (Chettih et al., 2023). We therefore predict that the dimensionality of the population code along the sensory hierarchy should decrease to support the learning of invariant sensory representations (Froudarakis et al., 2020), and then increase sharply at the apex, in the hippocampus, to enable the formation of detailed memories based on the specific combination of such invariant representations. This role of the hippocampus aligns with our observation of mixed selectivity, i.e., neurons learning similar feature combinations (Figure 5c), which in turn has been proposed to enable high-dimensional representations important for higher cognition areas (Fusi et al., 2016; Bernardi et al., 2020).

**Circuit mechanisms underlying memory formation**   The surprising observation of zero-shot learning of place cells (Figure 4) suggests that the sparse autoencoders learned to cluster similar samples onto single neurons in a generic manner. We hypothesize that the responsible circuits might correspond to known hippocampal processes, mainly pattern completion and pattern separation (Rolls, 2013). On the one hand, neurons are pushed to collapse across-sample variability by clustering samples based on similarity, an effect akin to pattern completion. On the other hand, sparsity also imposes sharp discontinuities between clusters in neuronal space, even when they might be close in input space, a process akin to pattern separation. The combination of both processes is reminiscent of the locality-sensitive hashing (LSH) algorithms used in the computer science field for fast image search (Kulis & Grauman, 2009). Future work will shed light on the learned circuit mechanisms behind such a LSH in sparse autoencoders, and their potential mapping to pattern separation and completion.

**Place cell over-representation near reward areas**   We have shown that the development of localized receptive fields can be generalized to other input modalities, such as sound. Crucially, we used this simplified framework to demonstrate that receptive field distribution tends to be modulated by the input sampling distribution (Figure 5d). Importantly, it has been observed that the density of place fields increases near reward areas (Mamad et al., 2017). This has led researchers to seek external reward signals in the hippocampus that would modulate the place cell map (Kaufman et al., 2020). However, based on our results, we propose an alternative explanation based on oversampling: animals tend to spend more time within rewarded areas due to consummatory behaviors, hence biasing sensory sampling and learning. Additionally, in line with this hypothesis, place cell trajectories leading to rewards or goals tend to be replayed more often than unrewarded past trajectories (Ambrose et al., 2016), which would further reinforce the sampling bias.

**Biological plausibility**   Although it has been claimed that error backpropagation and gradient descent are mechanisms that could be implemented in the brain (Lillicrap et al., 2020), particularly in the hippocampus (Santos-Pata et al., 2021b), we believe that such strong assumptions are unnecessary to map our model and observations to the real hippocampus. The orthonormal activity regularization term used in our sparse autoencoders could be realized by combining strong lateral inhibition (promoting pairwise decorrelation) and homeostatic plasticity (ensuring that neurons maintain equalized firing rates over time). Additionally, the orthonormal term can be thought of as sparse whitening

in ReLU-like neurons (i.e., pairwise decorrelation and variance normalization in the low-firing rate regime), a mechanism proposed to be realized in the brain by an overcomplete basis of inhibitory interneuron projections (Duong et al., 2023b;a; Lipshutz et al., 2022). Therefore, we contemplate several alternative mechanisms whereby the main objective driving our sparse autoencoders could be realized in brain circuits.

**Limitations**   While our reinforcement learning experiments suggest that DQNs can make use of very high-dimensional representations to solve complex tasks, the present study is limited in scope (Figure 6). We tested only a few tasks (four tasks within the Animal-AI testbed) and a single model (DQN). To gain a more comprehensive understanding of the suitability of hippocampal-like representations for behavioral and policy learning, further testing is required with a broader range of tasks and models, especially in non-stationary environments where unseen samples are the norm. Additionally, future research should explore how these sparse autoencoders could enhance reinforcement learning algorithms that rely on discrete representations, potentially enabling algorithms based on, e.g., the successor representations (Dayan, 1993), to extend beyond simplified grid worlds.

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

# A APPENDIX

## A.1 DETAILED METHODS

### A.1.1 MODEL'S ARCHITECTURE AND TRAINING

**Visual autoencoder (Vis-AE)** To compress the images from the Animal-AI enviroment, we employed a convolutional autoencoder (ConvAE) that maps the $3 \times 84 \times 84$ to a latent space with 1000 neurons, effectively compressing the images by a factor of $\sim$21. The details on the architecture can be found in Table 1.

Table 1: Architecture of the Visual Autoencoder (Vis-AE)

| Layer | Type | Act. Func. | Filters/Units | Kernel Size | Stride/Padding |
|-------|------|------------|---------------|-------------|----------------|
| Input | Input | - | $3 \times 84 \times 84$ | - | - |
| Conv1 | 2D Conv. | ReLU | 16 | $4 \times 4$ | 2/1 |
| Conv2 | 2D Conv. | ReLU | 32 | $4 \times 4$ | 2/1 |
| Conv3 | 2D Conv. | ReLU | 64 | $4 \times 4$ | 2/1 |
| Reshape | Reshape | - | 6400 | - | - |
| Fc1 ($Z$) | Linear | ReLU | 1000 | - | - |
| Fc2 | Linear | ReLU | 6400 | - | - |
| Conv4 | Trans. 2D Conv. | ReLU | 32 | $4 \times 4$ | 2/1 (out. pad. 1) |
| Conv5 | Trans. 2D Conv. | ReLU | 16 | $4 \times 4$ | 2/1 |
| Conv6 | Trans. 2D Conv. | Sigmoid | 3 | $4 \times 4$ | 2/1 |
| Output | Output | - | $3 \times 84 \times 84$ | - | - |

**Audio autoencoder (Aud-AE)** To compress the frequency signals from the synthetic sound dataset, we employed another ConvAE that maps one-second time series data (with a resolution of $10^{-3}$ s) to a latent space with 100 neurons, effectively compressing the signals by a factor of 10. The details on the architecture can be found in Table 2.

Table 2: Architecture of the Audio Autoencoder (Aud-AE)

| Layer | Type | Act. Func. | Filters/Units | Kernel Size | Stride/Padding |
|-------|------|------------|---------------|-------------|----------------|
| Input | Input | - | $1 \times 1000$ | - | - |
| Conv1 | 1D Conv. | ReLU | 16 | 100 | 2/1 |
| Conv2 | 1D Conv. | ReLU | 32 | 100 | 2/1 |
| Conv3 | 1D Conv. | ReLU | 64 | 100 | 2/1 |
| Reshape | Reshape | - | 2624 | - | - |
| Fc1 ($Z$) | Linear | ReLU | 100 | - | - |
| Fc2 | Linear | ReLU | 2624 | - | - |
| Conv4 | Trans. 1D Conv. | ReLU | 32 | 100 | 2/1 |
| Conv5 | Trans. 1D Conv. | ReLU | 16 | 100 | 2/1 |
| Conv6 | Trans. 1D Conv. | Tanh | 1 | 100 | 2/1 |
| Output | Output | - | $1 \times 1000$ | - | - |

**Loss function** The loss function to be minimized in both autoencoders (Vis-AE and Aud-AE) includes a mean squared error (MSE) term as a reconstruction error to force the latent space to preserve the input information, and a orthonormal activity regularization term that promotes sparse representations in the latent space $Z$:

$$\mathcal{L} = \frac{1}{m}\|\mathbf{X} - \hat{\mathbf{X}}\|_2^2 + \frac{\lambda}{mn}\|\mathbf{I}_n - \mathbf{Z}^{\mathrm{T}}\mathbf{Z}\|_{\mathrm{F}}, \tag{2}$$

where $\mathbf{X}$ is the input, $\hat{\mathbf{X}}$ is the output (i.e., the reconstructed input), $\lambda$ is the regularization coefficient ($10^3$ by default), $\mathbf{I}_n$ is the identity matrix with shape $n \times n$, $\mathbf{Z}$ is the middle layer's activity matrix of shape $m \times n$, $m$ being the batch size, and $n$ the number of hidden units. Therefore, the Gramian $\mathbf{Z}^{\mathrm{T}}\mathbf{Z}$

(with shape $n \times n$) captures the pairwise co-activation strengths between neurons in latent space. The symbols $\|\cdot\|_2$ and $\|\cdot\|_F$ denote the squared L2-norm and the Frobenius norm, respectively. The orthonormal activity regularization term promotes pairwise neuron decorrelation while achieving an equalized contribution across neurons, alleviating the dying ReLU problem. For comparisons with the dense AE, $\lambda$ was simply set to 0. We have found this orthonormal activity regularization to give improved and more reliable results than the L1 activity regulation term used in standard sparse autoencoders, especially in preventing the dead ReLU problem.

**Data generation and training of Vis-AE** For the visual experiments, datasets were generated by sampling a total of 10000 images in each of four Animal-AI environments: "doubleTmaze", "permanence", "cylinder", and "thorndike", at random locations within the arena (excluding the 10% of the space closest to each wall) and with random angles, following uniform distributions. Training was conducted using batches of 256 images for 10000 epochs, with independent training runs per environment. The Adam optimizer with no weight decay was used to train the network and the learning rate was set at $10^{-4}$. In addition, the regularization strength $\lambda$ was set to $10^3$ for the sparse autoencoder, and 0 for the dense autoencoder. All weights were initialized using Xavier initialization except for $Z$ (i.e., Fc1), whose weights were initialized following a random asymmetric initialization to minimize the dying ReLU problem (Lu et al., 2019). An early stop of 0.0005 in reconstruction loss was used to compare sparse and dense autoencoders with similar reconstruction capabilities.

**Data generation and training of Aud-AE** For the synthetic audio experiments, training consisted of batches of 256 one-second audio slices of varying frequencies. A sliding window of 1 second (with 1 ms shift) was applied to a linearly-varying frequency signal of total time 100 seconds, moving from 10 to 80 Hz, hence resulting in a total of 99001 samples. The sampling frequency was set at $10^4$ Hz, so that the kernel size (1000) matched to one full cycle at the lowest input frequency (10 Hz). Training was conducted for 1000 epochs using the Adam optimizer with a learning rate of $10^{-4}$, with no weight decay. Here, the regularization strength $\lambda$ was set to $10^4$ for the sparse autoencoder and 0 for the dense autoencoder. The weights were initialized as with the visual-AE, with Xavier initialization in all layers except for the the latent space $Z$ that followed a random asymmetric initialization. An early stop of 0.002 in reconstruction loss was used to have a fair comparison between sparse and dense autoencoders.

A.1.2 SPATIAL TUNING

**Firing ratemaps** To generate ratemaps from latent space activity, we first created a grid of $60 \times 60$ bins (or $30 \times 30$ for computing spatial information scores) for each neuron. For each bin in the grid, we summed the neuron's activity values for images sampled within that bin, generating an activity map in space. Then, an occupancy map was generated to account for the variability in the number of images sampled at each spatial bin (sampling density), which was used to normalize the values in the activity map. Finally, Gaussian smoothing was applied to each neuron's normalized activity map, using a standard deviation of 3 bins. The resulting maps were normalized to their corresponding maximum values, yielding smooth ratemaps representing spatially-distributed neural activity.

**Place field identification** To identify and quantify place fields in each neuron's ratemap, we first binarized the ratemap by setting pixels with activity below 20% of the maximum activity to zero (inactive bins) and those above to one (active bins). Clusters were identified by grouping adjacent active bins, forming a cluster if a group of active bins was completely surrounded by inactive bins. Clusters not meeting the size criteria for place cells (between 3% and 50% of the total number of bins, 3600) were discarded. The remaining clusters were considered place fields.

**Spatial information** Spatial information (SI) scores measure the amount of information a neuron's firing rate ($\nu$) conveys about the agent's position ($\mathbf{r}$). For each neuron, we first normalized its ratemap (using a $30 \times 30$ bin grid) by the overall mean activity $\bar{\nu}$. Then, we computed an occupancy map that was normalized by the total number of samples to reflect the proportion of "time" spent in each bin of the ratemap, denoted as $p(\mathbf{r})$. Finally, we applied the formula introduced in Skaggs et al. (1992) to compute the SI scores:

$$\text{SI} = \sum_{\mathbf{r} \in \mathbf{R}} \frac{\nu(\mathbf{r})}{\bar{\nu}} \log_2 \left( \frac{\nu(\mathbf{r})}{\bar{\nu}} \right) p(\mathbf{r}). \tag{3}$$

The average SI across all neurons in the latent space $Z$ provides an estimate of the degree of spatial tuning that the model has developed.

**Spatial position decoding**  The spatial decoding error measures the expected error of a linear decoder using latent space activations $Z$ to predict the spatial position $\mathbf{r}$. We fit a linear regression model with $Z$ as the independent variables and $\mathbf{r}$ as the dependent variables, predicting positions as $\hat{\mathbf{R}} = Z\mathbf{W}$. Then, we compute the mean squared error (MSE) between the predicted positions $\hat{\mathbf{R}}$ and the actual ones $\mathbf{R}$:

$$\text{MSE} = \frac{1}{n_{\text{samples}}} \|\mathbf{R} - \hat{\mathbf{R}}\|_2^2. \tag{4}$$

Finally, the average spatial decoding error (MSE) is re-scaled by dividing it by the maximum distance in the environment, that is, the diagonal of the arena, computed as $d = s\sqrt{2}$, with $s$ being the side length.

### A.1.3 INTERPRETABILITY

**Visualizing and quantifying the network's tiling of the image space**  We employed the CLIP neural network (Radford et al., 2021) to encode images (resized from $3 \times 84 \times 84$ to $3 \times 224 \times 224$) into 512-dimensional vectors. These vectors were subsequently reduced to a two-dimensional representation using UMAP (McInnes et al., 2018), with 10 neighbors and a minimum distance of 0.1, enabling the visualization of the high-dimensional image space.

Neurons in the hidden layer of the autoencoder that exhibited strong activation in response to specific images—those triggering activations exceeding a certain % of their maximum activation across the dataset—were mapped to points in the 2D image space. We then identify clusters of points using the DBSCAN algorithm (Ester et al., 1996), with radius $\epsilon$ of 1 and minimum samples of 4. These parameters are very dataset-dependent and were thus selected and validated via extensive visual inspection to ensure reliable cluster identification. Convex hulls were constructed around these clusters using the Quickhull algorithm (Barber et al., 1996) to delineate their spatial boundaries. This allowed us to identify the regions of the input space that each neuron encodes in their activations, i.e., their receptive fields.

Let $\{H_i\}$ denote the set of convex hulls corresponding to each neuron's activated image space. The average overlap metric, $\overline{O}$, was calculated as follows:

$$\overline{O} = \frac{1}{\binom{k}{2}} \sum_{i<j} \frac{\text{Area}(H_i \cap H_j)}{\text{Area}(H_i \cup H_j)}, \tag{5}$$

where $\text{Area}(H_i \cap H_j)$ represents the area of intersection between hulls $H_i$ and $H_j$, $\text{Area}(H_i \cup H_j)$ is the area of the union of hulls $H_i$ and $H_j$, and $k$ is the total number of hulls. The hull calculations were performed using the Shapely Python library (Gillies, 2013). The metric $\overline{O}$ thus represents the average proportion of overlap relative to the union for each pair of hulls and ranges from 0 (no overlap) to 1 (complete overlap), thereby providing a quantitative measure of the redundancy in the neurons' receptive fields across the image space.

**Neuron clamping and decoding**  To test whether neurons in $Z$ were directly interpretable based on their single-neuron activity (therefore obviating population codes), we conducted clamping experiments. This involved setting the activation of a specific neuron $i$ in $Z$ to its maximum activation value observed across the dataset $\mathcal{X}$, while setting the activations of all other neurons to zero. This is represented as $z_i' = (0, \ldots, 0, x_{\max}, 0, \ldots, 0)$ where $x_{\max} = \max(\{z_i | z = f(x), x \in \mathcal{X}\})$ and $f(x)$ represents the encoding function mapping $\mathcal{X}$ to $z$. Then, $z_i'$ is processed by the decoder $g(z_i')$ (with $g(x)$ representing the decoding function mapping $z$ to $\hat{\mathcal{X}}$) to yield an output signal (image of audio wave, depending on the AE).

**Population code dimensionality**  The dimensionality of the population code was estimated by computing the power-law exponent $\alpha$ of the latent space activity $Z$ (Stringer et al., 2019). We performed PCA on $Z$ and computed the linear fit of the resulting eigenspectrum in log-log space over the range of the first 10 to 100 principal components. Since the exponent $\alpha$ provides an estimate of how fast the population activity eigenspectrum decays as new dimensions are added, high $\alpha$ values are indicative of low-dimensional codes, whereas low $\alpha$ values indicate high-dimensional codes.

### A.1.4 REINFORCEMENT LEARNING EXPERIMENTS

**Animal-AI Testbed**    The Animal-AI testbed is a comprehensive platform designed for evaluating the cognitive and learning capabilities of AI agents in a variety of tasks that simulate real-world challenges (Beyret et al., 2019). This testbed provides diverse environments where agents must use visual cues and navigate complex structures to achieve specific goals. The visual inputs from these environments are standardized to a resolution of 84 by 84 pixels, and agents can perform actions defined by a 2-dimensional vector of integers: the first component goes from 0 to 2 and corresponds to not moving, moving forward, or moving backwards, respectively; and the second component also goes from 0 to 2 and corresponds to not rotate, rotate left, or rotate right, respectively. To encourage efficient behavior, a standard frameskip of 4 is applied, and the reward value decreases by 0.001 at each step. Episodes terminate either when the agent obtains the reward or after 1000 frames.

We evaluate our reinforcement learning agents using four distinct benchmarks within the Animal-AI testbed: the Double T-maze, Object Permanence, Cylinder, and Thorndike tasks. Each of these tasks presents unique challenges that require the agent to apply different strategies and cognitive abilities.

- **Double T-maze**. Each episode starts with the agent positioned randomly at one of the corners of the maze, and the objective is to navigate to the center to obtain the reward. The center contains the only positive reward (+3) available in the environment. Due to the high and opaque maze walls, the agent cannot directly see the reward and must explore the maze to find it.

- **Object Permanence**. At the beginning of each episode, the agent observes a large reward (+3) falling behind a wall until it is completely occluded. The agent must then navigate to the hidden reward, avoiding a small and visible reward (+1) along the way.

- **Cylinder**. This task involves an opaque cylinder with a medium-sized reward hidden inside. The agent begins outside the cylinder and must navigate into the cylinder to obtain the reward (+2).

- **Thorndike**. The task tests the agent's ability to escape from a closed box to reach a reward located outside the box. The box is semi-transparent, allowing the agent to see the reward from inside. The only exit is blocked by a movable obstacle that the agent must push to escape. A medium reward (+2) outside the box is the sole positive reward available.

**Model**    To evaluate the performance of our sparse autoencoders in reinforcement learning scenarios, we used a standard Deep Q-Network (DQN) architecture (Mnih et al., 2015) with modifications to the input layer. Instead of feeding raw pixel data from the Animal-AI environments, we used the compressed representations of 1000 units generated by the Visual Autoencoder (Vis-AE).

The loss function optimized by the Deep Q-Network (DQN) is the Mean Squared Error (MSE) between the predicted Q-values and the target Q-values, calculated using the Bellman equation:

$$\mathcal{L}(\theta) = \mathbb{E}_{(s,a,r,s',d)\sim\text{ReplayBuffer}} \left[ \left( r + \gamma \cdot (1 - d) \cdot \max_{a'} Q_{\text{target}}(s', a'; \theta^-) - Q_{\text{main}}(s, a; \theta) \right)^2 \right] \quad (6)$$

where $Q_{\text{main}}$ is the main Q-network with parameters $\theta$, $Q_{\text{target}}$ is the target Q-network with parameters $\theta^-$, $s$ is the current state, $a$ is the action taken, $r$ is the reward received, $s'$ is the next state, $d$ is a boolean indicating whether $s'$ is a terminal state, and $\gamma$ is the discount factor. This loss function aims to minimize the difference between the Q-value predicted by the main network and the target Q-value, which is computed based on the reward and the maximum Q-value of the next state predicted by the target network. The training of the DQN was performed by using the RMSprop optimizer. The target network was periodically updated with the weights of the main DQN to stabilize training. The DQN was trained with the following hyperparameters: a learning rate of 0.00025, a discount factor ($\gamma$) of 0.99, an update frequency of 4 steps, and a target network update frequency of 2500 steps. The $\epsilon$ for the epsilon-greedy policy started at 1 and decayed linearly to 0.1 over 25000 steps. The replay buffer size was set to 25000, with a batch size of 32 for experience replay. The details on the architecture can be found in Table 3.

**Training and performance metrics**    Each reported experiment tested two DQN agents, Sparse and Dense, which differ only in their use of different Vis-AE models (sparse and dense autoencoders, respectively) to obtain compressed representations from the environment observations as input. The

Table 3: Architecture of the Deep Q-Network (DQN)

| Layer | Type | Act. Func. | Units |
|--------|--------|------------|-------|
| Input | Input | - | 1000 |
| Fc1 | Linear | ReLU | 100 |
| Fc2 | Linear | ReLU | 50 |
| Fc3 | Linear | ReLU | 25 |
| Fc4 | Linear | ReLU | 9 |
| Output | Output | - | 9 |

two agents were evaluated across the four Animal-AI tasks described earlier. Each model run lasted 5000 episodes, and to ensure statistical reliability, each model played each task between 20 and 27 times. The reported average performance metric was calculated using a sliding window of 20 episodes.

