# OpenReview forum: "Discretization of continuous input spaces in the hippocampal autoencoder"
_ICLR.cc/2025/Conference — Submitted to ICLR 2025_

### Official Review · Reviewer_jZg7 · 2024-10-27

**Soundness:** 3
**Presentation:** 2
**Contribution:** 2
**Rating:** 5
**Confidence:** 4

**Summary:**

This paper demonstrates a sparse autoencoder-based modelling approach to understand the encoding mechanism of place cells. Through a variety of experiments on spatial navigation tasks with visual stimulus, sound frequency and reinforcement learning agents, the authors have demonstrated that sparse autoencoder can be used to explain the computational mechanism behind the discretization of continuous, high-dimensional inputs i.e., place cells in the hippocampus, and that memory encoding and place cells are closely linked.

**Strengths:**

* The authors have conducted a comprehensive set of experiments, including:
   * spatial navigation with visual inputs: I really appreciate the usage of Animal-AI environments for the experimentation, which distinguishes this paper with earlier ones like Benna and Fusi (2021) that only performed the experimentation on simulated tasks with low-dimensional inputs.
   * In-depth analysis of the latent structure of the autoencoder.
   * Investigation of frequency-sensitive “place cells”, which shows the generalizability of the model.
   * Reinforcement learning agents.
* The writing is clear and easy to follow.
* There is no obvious error in the paper.

**Weaknesses:**

* Main contribution is unclear:
  * Despite the comprehensive experiments presented in this work, I found the main contribution poorly defined, especially given that earlier works (e.g., Benna and Fusi 2021, Santos-Pata et al. 2021) have followed similar autoencoder-based approaches.
  * For example, the authors stated in the final sentence of abstract that the findings demonstrate how sparsity gives rise to interpretable memories and thus establishes a link between memory and place cells. However, it has been shown in multiple models that sparsity is a necessary condition for obtaining place cells, regardless of the model (e.g. Chen et al., 2024, Benna and Fusi, 2021), and the second part of the claim is exactly what has been studied in Benna and Fusi 2021. Given these, what is the novel contribution of this paper?
  * One possible contribution that the authors can highlight is that the paper provides a full-scale investigation of 1) the necessity of sparsity and 2) the relationship between memory and space representation with experiments on higher-dimensional inputs of different modalities. However, this is only my opinion, and the authors may have other contributions they wish to highlight. In that case, please make sure they are sufficiently highlighted in the paper.
  * Related to this, I feel the paper could be improved by having a ‘Related Work’ section, which highlights what has been done and what hasn’t been done in earlier literature, and how this paper addressed these gaps.
* Lack of key experimental details in the main text:
  * I agree many experimental details (e.g., training information like learning rates) don’t have to be included in the main text. However, this paper has missing experimental details that affects understanding. Here’s an incomplete list of them:
    * Section 2.1: I hope the authors can use a few sentences to describe what exactly does the agent do in these 4 animal-ai environments, or what exactly the task is, as some readers might be unfamiliar with the dataset.
    * Section 2.2: In the first sentence of the second paragraph the authors mentioned “embedding of all samples” – as a reader I’m not sure what all samples refers to. I assume they are the images sampled from the tasks? However, which task did you use to get the samples? One of them or all? Also, although CLIP, UMAP, DBSCAN are well-known embedding/dim-reduction methods, you should cite relevant literature.
    * Section 2.4: The authors mentioned that the autoencoder is trained on one environment and tested on all others. Which environment is the training one and which are the testing ones? Did you do a cross-validation like testing? If not, I think it is preferable to do so as it will demonstrate the robustness of the results better.
    * Section 2.5: I struggle to understand the purpose of Fig 5d and the sentence associated with it: why do we need to bias the sampling towards certain frequencies?
    * Section 2.6: The authors moved the whole task description of these RL experiments to the Appendix, so it is quite unclear to me what exactly is the performance in Fig 6b measured on. It would be much nicer to include in the main text what the tasks and metrics are.

* Redundant experiments: I suggested in the previous point that the paper could benefit from adding more experimental details. This might be challenging given the limited number of pages. However, I believe there are some redundant experiments in this paper that can be removed to give space to more details for other, more informative experiments. For example, I didn't find the results in Fig 3 very unexpected/intriguing. Since orthogonality constraint is imposed on the autoencoder, it is unsurprising that the eigenspectrum of the sparse autoencoder is a flat line, because it is exactly what orthogonality means i.e., neurons will be more independent and explain equal amounts of variance in the data. I understand that the authors wanted to reproduce the results with mice in Stringer et al. 2019 but keep in mind the causality of the results: in mice, the eigenspectra reveal a coding mechanism, whereas in autoencoders, the coding mechanism (orthogonality/sparsity) is first imposed so the observed eigenspectra are expected. I’m wondering if the authors are willing to remove this experiment, or simply move it to the Appendix.

Overall, given the final two points above, I suggest the authors to think about the allocation of paper content on different experiments more carefully. Specifically, they may want to consider removing contents related to Fig3 and add more details to other experiments.

**Questions:**

* Abstract: I hope the authors could shorten the abstract and extract important message they want to deliver in this paper. Currently, the abstract is a pretty detailed summary of all the experiments. I don’t feel this level of details is necessary for the abstract.
* I wonder if the results (e.g., place fields) in the experiments are a function of \lambda? For example with a smaller \lambda i.e., weaker sparsity constraint, will the rate maps become more band-like? You may want to include these as appendix, just to show the sensitivity of the results to a quite important hyperparameter.
* I found the results in Fig 2d on memory quite interesting. I wonder if the authors have come across Radhakrishnan et al. 2020 that shows that autoencoders are associative memory models (like Hopfield Nets). This may offer you an interpretation of your results that in sparse autoencoders, individual neurons are basically “attractor neurons”.
* Is it possible to map the sparse autoencoders to the hippocampal circuit? For example, dentate gyrus is known to perform some kind of orthogonalization. How can it be related to the circuit mechanism implied by the sparse autoencoder?

Overall, I will suggest a score 5 to this paper. However, if they authos can address the concerns raised above, particularly on highlighting the contribution better, I'm happy to raise it to 6.

References:
Radhakrishnan et al. 2020: Overparameterized neural networks implement associative memory

---

> ### Author Response · Authors · 2024-11-24
>
> We thank reviewer jZg7 for their thoughtful comments and suggestions. Below, we address the weaknesses and questions raised and outline our revisions.
>
> Weaknesses.
> 1)
> We acknowledge the need to better articulate our unique contribution, particularly in light of related work. While Benna and Fusi (2021) already argued that place cells are memory cells, our work identifies the specific visual encoding properties—namely, contractive visual encoding—that produce spatial tuning. That is, nearby locations produce visually similar images, which cluster onto the same neuron, resulting in a certain degree of invariance in image space. When activity is then correlated with spatial positions, neurons are said to exhibit "spatial selectivity." Importantly, this is achieved without path-dependent processes (we sample randomly), and using high-dimensional inputs (images). In addition, we extend these findings to the auditory domain (e.g., Aronov et al., 2017) and demonstrate that such representations can be used even by standard DQN, despite their sparse and discrete-like nature. To emphasize our contributions, we will revise the abstract, main text, and add a Related Work section that explicitly highlights the similarities and differences with prior studies.
>
> 2)
> - Although the 4 tasks are briefly described in paragraph "Animal-AI Testbed" (section "A.1.4. RL experiments"), we will expand on the description of the four Animal-AI tasks in the main text and include supplementary videos of agents performing the tasks.
> - We will clarify that all training and analyses are performed separately for each environment, with pooled statistics used in certain cases (e.g., Figure 1c, d). For Figure 2, panel a and b are from the Cylinder task (see caption), while panel c pools results across tasks. These details will be more explicitly stated throughout the text to avoid confusion. Additionally, we will cite the relevant literature for CLIP, UMAP, and DBSCAN.
> - It is indeed a cross-validation-like testing, where each environment is tested against all others, and then we report the pooled statistics. We will clarify this in the main text.
> - In Figure 5d we show that the allocation of place fields follows the distribution of the input: when it is uniform, the mapping is linear, whereas when it is a Gaussian the allocation (and width) of the place fields changes accordingly. This is not only relevant to show as a general property of the model, but it also provides an alternative explanation for the place cell over-representation in reward areas (as explained in the Discussion).
> - The average reward metric used in RL experiments will be better contextualized, particularly regarding task-specific maximum rewards.
>
> 3)
> While flat eigenspectra for sparse autoencoders are expected due to orthonormal regularization, the eigenspectra for dense autoencoders showing decay near 1 is not, highlighting an important contrast. Panels b and c of Figure 3 provide additional insights: the breaking of input-output similarity (previously deemed necessary for localized receptive fields) and the connection between non-orthogonal kernels and orthogonal activity, linking high-dimensional representations to mixed selectivity. Nevertheless, as these figures are less central to our main message, we will move them to the supplementary material for clarity.
>
> Questions.
> 1) We agree with the reviewer and will revise the abstract to emphasize the main contribution: linking contractive visual encoding to the emergence of place cells.
> 2) We will include a supplementary figure showing how metrics like spatial information vary with α. As we have observed in prior experiments, the transition is rather sharp, and we agree that this result is actually important to report.
> 3) We are grateful for the reference provided by the reviewer, which offers a valuable explanatory framework for our results. We will cite and discuss this work in the Discussion, along with its implications for future research, such as linking the partitioning of the image space (Figure 2a and b) to the identification of basins of attraction in the image space through iterative decoding.
> 4) We think that the main components of our orthonormalization term are consistent with known hippocampal mechanisms:
> - Decorrelation (off-diagonal SSCP matrix (Z.T*Z) elements → 0) aligns with SST-based mutual inhibition in DG or CA1.
> - Activity equalization (diagonal SSCP elements → 1) reflects slow homeostatic plasticity (keeping firing rates fairly constant in the long term).
> - Feature learning mirrors Hebbian or STDP-like learning processes.
> We will expand on this in the Discussion.
>
> Overall, we agree with the reviewer in that the text needs to be modified to emphasize the main contributions of the work and clarify important details about the methods. We believe these revisions will address the reviewer’s concerns and improve the paper’s clarity.

---

### Official Review · Reviewer_ab1v · 2024-11-01

**Soundness:** 1
**Presentation:** 4
**Contribution:** 3
**Rating:** 3
**Confidence:** 5

**Summary:**

This paper investigates how sparse autoencoders, equipped with orthonormal activity regularization, can simulate place cell-like activity in artificial neural networks, potentially providing insights into hippocampal encoding mechanisms. Through comparisons between dense and sparse autoencoders trained on images of 3D environments, the authors demonstrate that sparse autoencoders generate discrete, interpretable representations by mapping similar inputs to localized receptive fields with minimal overlap. This mechanism is similar to locality-sensitive hash, creating high-dimensional representations that resemble spatial maps, similar to place cells in the hippocampus. Additionally, the study generalizes these findings to the auditory domain, showing that sparse autoencoders can produce frequency-specific receptive fields, suggesting modality-independent encoding.

The authors introduce a novel metric to quantify the discretization of image space and present results that reinforcement learning agents benefit from these high-dimensional sparse representations in some environments. The paper concludes with a discussion on the biological plausibility of orthonormal regularization, suggesting it could be implemented through neural processes like lateral inhibition and homeostatic plasticity. Overall, the work bridges computational neuroscience and artificial intelligence by proposing a model where sparse input compression supports episodic memory formation and spatial representation in a hippocampal-inspired framework.

**Strengths:**

#### **Originality**
This paper introduces an innovative approach by using sparse autoencoders with orthonormal regularization to simulate place cell-like behavior, bridging neuroscience and machine learning. This combination of techniques is a creative application in hippocampal modeling, especially as it applies across sensory modalities (visual and auditory) and introduces a new metric for discretization of input spaces.

#### **Quality**
The experimental design is thorough and well-documented, including comparisons across dense and sparse models, detailed model architecture, and training procedures. The experiments are contextualized within neuroscience theories like pattern separation/completion, lending scientific depth and grounding to the study. However, while rigorous, the experiments are limited to specific, controlled environments, which may impact generalizability.

#### **Clarity**
The paper is generally clear in explaining complex ideas, with visuals that effectively illustrate neuron activity maps and spatial encoding properties. The appendix offers additional transparency regarding methods, adding to reproducibility. Nonetheless, some technical details—such as orthonormal regularization’s biological plausibility—could be expanded to improve accessibility.

#### **Significance**
This work has notable implications for both neuroscience and AI, providing a potential model for hippocampal-like representation and spatial coding that might inform further studies in spatial cognition. Its cross-modal approach (visual and auditory) hints at a generalizable model for sensory representation, though further validation across a broader range of tasks and sensory modalities would strengthen its practical impact.

**Weaknesses:**

### **Inaccurate use of terms**
The paper’s use of terms like "memory" and "locality-sensitive hashing (LSH)" misrepresent its functionality, suggesting capabilities that the model doesn’t fully achieve.

The term "memory" implies persistence, flexible retrieval, and time-based storage. Episodic memories, in particular, include context, time, and location (what, when, where). The model, however, lacks mechanisms for retaining and recalling information across time, relying instead on transient representations accessed by direct activation (neuron clamping). Referring to these as "representations" or "spatial encodings" would more accurately reflect their function.

LSH typically involves hashing for efficient, similarity-based retrieval, using probabilistic mappings to preserve approximate similarity and enable efficient search. The model’s rigid clustering lacks LSH’s probabilistic flexibility and retrieval efficiency, as it does not include true hashing or indexing. Describing the model as “localized spatial encoding” or “input-space tiling” would avoid implying LSH-like efficiency or robustness.

__Suggestion__: Replace "memory" with "representation" and "LSH" with “localized encoding” to better match the model’s capabilities. If these terms are retained, clearly specify that the model only partially mimics these concepts. Compare to methods like receptive field mapping or population coding for a more accurate portrayal.

### **zero-shot learning**
The authors claim that sparse autoencoders exhibit zero-shot learning of place cells across novel environments, but there is a limited exploration of why or how this occurs. The model’s capacity for generalization appears promising but has not been properly examined. Analyzing the network's internal structure or activity patterns when exposed to novel environments could reveal mechanisms that enable this generalization. Running ablation studies on the architecture (e.g., removing the orthonormal regularization) or visualizing the learned feature space across environments would provide insights into which aspects drive the process. Also, how does the decoding look like during zero-shot learning?

Moreover, when the model encounters unseen data, interpreting its representations as "memory" becomes challenging, as true memory implies both stored information and the ability to generalize or recall similar contexts flexibly. When the model encounters unseen data, it generates new, sparse representations in the latent space without direct prior storage or memory of these specific inputs. This process resembles on-the-fly encoding rather than recall from stored experiences, suggesting that the model is generating new representations rather than accessing a pre-existing memory.

In zero-shot learning, the model's response to new data could reflect a form of pattern generalization based on similarities to previously encoded features. While this may resemble aspects of memory, it lacks associative recall or the ability to adapt previously stored information to novel contexts, a hallmark of true memory.

The representations formed for new inputs suggest flexible encoding rather than dynamic recall, as they are generated without modification of or access to previously stored states. This is akin to a flexible encoding mechanism rather than a memory system, as it adapts to new data but does not actively retrieve, compare, or contextualize it based on stored knowledge.

### **bio plausability and regularization**
The paper claims that sparse autoencoders with orthonormal regularization simulate place cell-like properties in the hippocampus, but the biological plausibility of this regularization remains speculative. Orthonormal regularization is uncommon in neural modeling and lacks a clear parallel with hippocampal processes.

Moreover, it is not yet clear if the place cell-like properties are specific to this regularization or could be achieved with alternative, potentially more biologically plausible methods.
Implementing and testing additional regularization strategies, such as L1 or L2 regularization combined with decorrelation constraints, could reveal whether place cell emergence depends specifically on orthonormal regularization. This would not only validate the robustness of the current approach but also explore alternative mechanisms that might be more feasible in biological systems. There are several propositions in the literature see e.g. [Schaeffer et al.](https://arxiv.org/abs/2311.02316)

### **Interpretabilty**
The interpretability claims based on clamping neurons to their maximum observed activity are primarily qualitative, which may limit reproducibility and objectivity.
Including quantitative metrics—such as cosine similarity or structural similarity index (SSIM) scores—between the decoded outputs and training images would provide a more rigorous assessment of interpretability. If visualizations remain subjective, a standardized metric would enhance reproducibility and add credibility to the interpretability claims.

### **Sensory modalities**
While the authors extend the model to the auditory domain, the tested signals are relatively simple (frequency alone), which limits understanding of the model’s adaptability to more complex auditory stimuli.
Testing with more complex auditory signals, such as temporal sequences or multi-dimensional auditory stimuli (e.g., varying intensity, background noise), would demonstrate if the model’s generalization holds for richer sensory environments.  Moreover, the study from Aronov focused on a goal-directed task.

### **RL**
The paper’s reinforcement learning (RL) results show that sparse autoencoders perform inconsistently, which the authors attribute to the brittleness of high-dimensional representations. However, the exact limitations and causes of brittleness are not thoroughly analyzed.

A more detailed analysis of the conditions under which sparse representations lead to brittle behavior could provide actionable insights for improving stability. Testing across a wider range of RL architectures (beyond DQN) or incorporating additional stability measures, such as other regularization techniques, dropout, or noise injection, could yield strategies to mitigate brittleness and enhance adaptability in RL tasks.

**Questions:**

In general, I liked the paper and applaud the innovative use of autoencoders and the clarity of the authors’ representations (pun intended). However, the biological relevance and the application of terms like ‘memory’ and ‘LSH’ feel overstated, making the results overly speculative. If the claims were substantially backed up with well-stated limitations or if terminology were revised to better match the model's demonstrated capabilities (e.g., ‘representation’ rather than ‘memory’) and the other stated weaknesses were addressed, I would be inclined to increase my score.

---

> ### Author Response · Authors · 2024-11-22
> **Will rephrase to avoid misleading terms and provide extra analyses when feasible**
>
> We thank Reviewer ab1v for their supporting words, critical feedback, and valuable insights. Below, we address the concerns raised and provide clarifications.
>
> Weaknesses.
> 1) "Inaccurate use of terms". We appreciate the reviewer’s concern about the precise use of terminology. The term "memory" has varying interpretations across fields—for instance, memorization as overfitting in deep learning, associative memories in Hopfield networks, or memory buffers in reinforcement learning. Therefore, its adequacy relies on properly setting the context of its use. Importantly, overparametrized autoencoders have been shown to store training examples as attractors, which has been interpreted as a Hopfield-like associative memory (Radhakrishnan et al., 2020, PNAS). We will clarify these contextual meanings in the manuscript to avoid misleading terms. Then, regarding the term "LSH," it was used as an analogy to highlight that similar images cluster onto the same latent unit, akin to mapping similar inputs to unique one-hot vectors (indexing). However, as this analogy may be misleading due to particularities of LSH, we will revise the text for accuracy and rephrase where necessary.
>
> 2) "Zero-shot learning". We agree that this aspect of the model warrants deeper analysis. The model likely clusters similar images into single units, possibly leveraging shared features across environments. However, note that some environments are vastly different (e.g., Double T-maze vs. Cylinder, Figure 6a). Zero-shot "spatial" learning is then demonstrated by: (1) nearly identical place field distributions between training and unseen environments (Figure 4a, b) and (2) decoding agent positions from latent activity (Figure 4c). Inspired by Radhakrishnan et al. (2020), we hypothesize that unseen datasets are partitioned based on the training dataset manifold, producing localized receptive fields. While clarifying the mechanism is nontrivial and left for future work, the observed rapid emergence of place cells seems to align with rodent studies. We will add these considerations to the text to clarify its limitations and possible implications.
>
> 3) "Bio plausibility and generalization". We acknowledge the need for further discussion on this topic. In previous experiments, we confirmed that neither L1 nor L2 generate place cells. However, latent space whitening combined with L1 regularization effectively approximates orthonormalization (since units are restricted to have non-negative values; ReLU). We will argue that the main components of our orthonormalization term are consistent with known hippocampal mechanisms:
> - Decorrelation (off-diagonal SSCP matrix (Z.T*Z) elements → 0) aligns with mutual inhibition in DG or CA1.
> - Activity equalization (diagonal SSCP elements → 1) reflects slow homeostatic plasticity.
> - Feature learning mirrors Hebbian or STDP-like learning processes.
> These mechanisms collectively could approximate the observed regularization effects. We will expand on this in the Discussion while leaving full modeling of these processes for future work.
>
> 4) "Interpretability". Visual inspection of decoded outputs highlights differences between sparse and dense autoencoders (e.g., Figure 5c). To strengthen this however, we will add supplementary plots quantifying image similarity between decoded and original data using cosine similarity and SSIM scores.
>
> 5) "Sensory modalities". The auditory task is simplified in the frequency space to align with prior work (e.g., Aronov) and to facilitate interpretability (mirroring linear track studies in rodents. This simplification allows us to clearly observe how input statistics shape receptive field allocation (e.g., Figure 5b). For instance, uniform input distributions lead to uniform field coverage (Figure 5b sorted), while Gaussian-distributed inputs bias receptive field density accordingly (Figure 5d). Extensions to more complex datasets (e.g., spoken words) would provide an extra, ecologically-valid test of these principles, without necessarily affecting the main conclusions.
>
> 6) "RL". High-dimensional sparse representations are often considered brittle and poor at generalization. However, our results show that these high-dimensional sparse representations perform nearly as effectively as dense, low-dimensional ones in spatial navigation tasks. While a detailed analysis of brittleness is beyond this paper's scope, we acknowledge limitations in our RL experiments (e.g., task diversity and architectural comparisons) and propose future work on hippocampally inspired RL models, such as episodic RL, to explore these issues further. Further testing with other models is not within our capabilities given time and compute constraints. Instead, we are conducting additional Animal-AI tests to enhance the reliability of the reported results.
>
> We hope these clarifications address the reviewer’s concerns and enhance the paper’s clarity. We appreciate the feedback and look forward to further discussion.

---

> > ### Comment · Reviewer_ab1v · 2024-11-26
> >
> > Thank you for the clarification.
> >
> > 1. I agree that memory is used widely in both AI and neuroscience, to the extent that it loses its meaning. However, episodic memory has a more narrow meaning, and I do not agree that your results align with this. To retain meaning in these terms I would suggest that you write a specific definition of memory, or episodic memory, or use a different word.
> >
> > Other than that I have no further comments

---

> > > ### Author Response · Authors · 2024-11-26
> > >
> > > Thank you for your response.
> > >
> > > We originally used the term "episodic memory" due to the well-established relationship between hippocampal place cell activity and episodic memory. Additionally, the broader use of this term in recent literature probably reflects a shift in the scientific community, suggesting that its original, narrower definition may no longer fully capture our evolving understanding of the brain--especially with the emergence of similar phenomena in new complex models like deep neural networks.
> > >
> > > That said, we understand that this usage may generate controversy or confusion. To address this, we are happy to mainly use the term "memory" instead of "episodic memory", and add a clarifying statement to the manuscript, such as: "Throughout the paper, we use the term memory to refer to the development of distinct activity patterns that can be readily used to recover specific training examples" (e.g., please compare the two training examples of neuron 9 in Fig2a with the neuron's decoded image after activity clamping in Fig2d). We hope this clarification resolves your concerns. Otherwise, we are open to receive further feedback on this.
> > >
> > > Finally, as you have only followed up on one of the six weaknesses previously raised, we kindly ask you to confirm whether our responses have adequately addressed the remaining five points or if additional clarifications/action would be needed.

---

### Official Review · Reviewer_pMQH · 2024-11-03

**Soundness:** 1
**Presentation:** 3
**Contribution:** 2
**Rating:** 3
**Confidence:** 4

**Summary:**

This paper models the formation of hippocampal-like spatial representations by training sparse autoencoders on visual images from 3D environments. By applying an orthonormal activity regularization to encourage sparsity, the authors observe place cell-like behavior, where certain neurons respond selectively to specific regions in the input space. They argue that this sparsity enables the model to discretize continuous spaces into distinct memory-like representations. The study also extends to audio data, showing similar localized tuning, and evaluates how these representations impact reinforcement learning agents. The paper argues that such sparse coding mechanisms could illuminate how episodic memory and spatial representations emerge in the hippocampus.

**Strengths:**

1. Interesting use of the Animal-AI environment to study how phenomena commonly found in biological brains (place cells) might emerge.
2. Paper is clearly written (minus a few missing experimental details)

**Weaknesses:**

1. Confounding Visual and Spatial Representations: The study compares sparse autoencoder representations derived from static visual images to hippocampal place cells, effectively confounding visual encoding with spatial encoding. The hippocampus integrates both sensory cues (such as vision) along with self-motion cues (vestibular, proprioceptive, and motor information). The analogy to hippocampal place cells is therefore weak, as the model solely captures visual features, which may not directly translate to spatial encoding in biological brains.
2. "Sparsity" Enforced by Design, Not Necessarily Task-Relevant: The network's sparse representations are enforced through an orthonormal regularization term, which directly promotes decorrelation (not sparsity!) in the latent space. This enforced structure is interesting but does not convincingly link the sparsity to improved performance on the core task of reconstructing visual inputs. The paper lacks an analysis on whether the sparse representations improve or impair reconstruction accuracy, raising questions about whether these representations are useful for encoding spatial features or are merely artifacts of the regularization. I think the paper should compare both dense (no penalty), decorrelated (the orthonormal penalty used in the paper), as well as sparse (l1 penalty) networks; and show the performance on the task (reconstruction error) for each, ideally for a range of regularization strengths (values for ). How do the results in the paper depend on either (a) the choice of penalty used or (b) the regularization penalty strength ()?
3. Unclear Image Sampling and Environmental Modeling: The training dataset appears to consist of randomly sampled images from a virtual environment, but the paper provides little detail on the spatial structure of these samples. If the images are not derived from a structured, agent-driven navigation through space, it is unclear how the autoencoder captures spatial rather than purely visual information. This choice weakens claims that the model learns spatial representations akin to the hippocampal encoding of navigated environments. More rigorous sampling or agent-driven exploration would be necessary to substantiate claims of spatial learning. This would also ideally include some understanding of how the results depended on the choice of agent-driven exploration used.
4. Inadequate Evaluation of Representation Utility: The paper’s core claims focus on the emergence of hippocampal-like representations, but the practical utility of these representations remains untested in terms of spatial or episodic memory tasks. While place cell-like activity is demonstrated in the autoencoder’s neurons, it is unclear if these representations are suitable for real-world tasks that would benefit from spatial memory, such as pathfinding or goal-directed behavior. The RL experiments suggest the opposite (that the penalty is harmful), if I understand them correctly.

**Questions:**

* How were the images sampled when training the autoencoder?
* Could you be more specific about what exactly didn't work with using an explicit sparsity penalty (l1 norm)?
* How should I think about multi-modal signals related to navigation (e.g. vestibular, proprioceptive, and motor information)? Is the paper suggesting they are irrelevant for generating place cells?

---

> ### Author Response · Authors · 2024-11-22
> **A feature, not a bug: Place cells emerge from visual encoding**
>
> We thank Reviewer pMQH for their constructive comments. Below, we address the concerns raised and clarify potential misunderstandings.
>
> Weaknesses.
> 1) What the reviewer perceives as a limitation is, in fact, a key insight of our study: visual inputs alone are sufficient to generate place cells, whereas self-motion cues are not. Furthermore, we trained the autoencoders with randomized image orders, disrupting the temporal correlations inherent in spatial exploration. These findings highlight that place cells can emerge independently of path-dependence processes, challenging prior studies. Importantly, we focus on the connection between visual and spatial representations—this is not a confounder but the central contribution of our work. We demonstrate how visual encoding naturally leads to spatially localized place fields, based on nearby locations' image similarity. Thus, we argue that this is a strength rather than a weakness. We will stress this point more clearly in the text, clarifying the contribution of our work.
>
> 2) The orthonormal regularization term promotes decorrelation and sparsity because ReLU activations prevent negative activity. This regularization operates on the Sum of Squares and Cross-Products (SSCP) matrix (Z.T*Z), not the covariance matrix, ensuring off-diagonal SSCP elements approach zero and diagonal elements approach 1, thereby decorrelating and sparsifying activity while mitigating dead ReLU issues. As stated in the Detailed Methods ("Loss"), both sparse and dense autoencoders achieve equivalent reconstruction loss, ensuring a fair comparison: "An early stop of 0.0005 in reconstruction loss was used to compare sparse and dense autoencoders with similar reconstruction capabilities."
>
> 3) We would like to emphasize here again the central finding of the paper: that visual encoding naturally leads to spatial representations. As detailed in the Methods section: "For the visual experiments, datasets were generated by sampling a total of 10000 images in each of four Animal-AI environments: 'doubleTmaze', 'permanence', 'cylinder', and 'thorndike', at random locations within the arena (excluding the 10% of the space closest to each wall) and with random angles, following uniform distributions". The emergence of structured spatial tuning from random image sequences is a significant result, not a bug or a limitation. While we acknowledge that exploring how sampling strategies (e.g., wall-following or thigmotaxis in rodents) impact place field distributions is an intriguing avenue, we address this partially in the simpler, auditory domain (Figure 5d). A more extensive spatial analysis, while valuable, is outside the scope of this work and is better suited for future research.
>
> 4) The four Animal-AI tasks are spatial navigation challenges, demonstrating that our representations are suitable for reinforcement learning (RL). However, the RL experiments aim to show that sparse representations perform comparably to dense ones, not to prove their superiority. Why would DQN agents perform better with sparse inputs anyways? However, sparse representations are often criticized as brittle or poor at generalizing to unseen data, yet our DQN agents with sparse inputs successfully solve tasks while encountering novel inputs. While we acknowledge limitations in the RL experiments (discussed in the manuscript), exploring architectures more closely aligned with hippocampal processes, such as episodic RL, represents an important direction for future work to show its potential and actual benefits. Lastly, we are currently running more tests on Animal-AI to strengthen our results.
>
> Questions.
> 1) As described in the Detailed Methods ("Data generation and training of Vis-AE"): "For the visual experiments, datasets were generated by sampling a total of 10000 images in each of four Animal-AI environments: 'doubleTmaze', 'permanence', 'cylinder', and 'thorndike', at random locations within the arena (excluding the 10% of the space closest to each wall) and with random angles, following uniform distributions".
>
> 2) We detailed in "Loss function" that orthonormal regularization improves results by alleviating dead ReLU issues, a limitation of the L1 penalty (we tested a wide range of strength values). Using L1 alone systematically resulted in most units being inactive, with the remaining units failing to exhibit place cell-like properties or localized receptive fields.
>
> 3) Our results suggest that multi-modal signals are unnecessary for place cell formation. However, it is likely that these signals modulate place cell expression during real-world navigation (e.g., speed-dependent firing rates). Our paper focuses exclusively on the emergence of place cells, not a comprehensive model of their behavior, and challenges prevailing assumptions that multi-modal integration is required.
>
> We hope this clarifies our contributions and addresses the reviewer’s concerns. We appreciate the feedback and welcome further discussion.

---

> > ### Comment · Reviewer_pMQH · 2024-11-23
> > **thanks for the clarification**
> >
> > Thanks for your response. I am still struggling with how to interpret the results given that images are presented in random order. If I understand correctly, the implication is that "place" cells naturally emerge in purely visual networks, when those images correspond to images randomly sampled from an environment?
> >
> > If so, what is the connection to the hippocampus, which _does_ integrate motion cues (e.g. vestibular and proprioceptive inputs) directly, whereas as it integrates higher level visual information but not visual information directly (ie. there are no direct connections from low-level visual areas to the hippocampus, as far as I know). On the flip side, how come visual cortex doesn't have place cells?

---

> > > ### Author Response · Authors · 2024-11-23
> > >
> > > Thank you for the follow-up comments.
> > >
> > > Random sampling is not required for place cells to emerge. Indeed, in other projects involving robot navigation, we also observe place cells using the same type of sparse autoencoders with on-the-fly image sampling and batch training. However, previous (autoencoder) models of hippocampal/place cell formation (e.g., Benna et al., 2021, PNAS) rely on some kind of path-dependent mechanisms, which we explicitly show to be unnecessary. Instead, our findings suggest that place cells emerge due to a contractive visual encoding process: nearby locations produce visually similar images, which are clustered onto the same neuron. Thus, the neurons shows a certain degree of invariance in image space. When experimenters (computational of biological) correlate the activity with spatial positions, the neurons appear to exhibit "spatial selectivity". While Benna et al. (2021) already argued that place cells are actually memory cells, our work identifies the specific visual encoding properties that drive spatial tuning, which is the main contribution of our study.
> > >
> > > The hippocampus is a brain structure known to sit at the apex of the sensory processing hierarchy. Therefore, it effectively "integrates" various sensory and cognitive signals (e.g., social, emotion, etc.), but their relevance to place cell formation remains unclear. The real question is then: how important are these signals in determining the formation of place cells within the hippocampus? We kindly invite the reviewer to provide references supporting the necessity of including such signals in our model of place cell formation. To our knowledge, vestibular and proprioceptive inputs primarily influence head direction and grid cells in the medial entorhinal cortex, while higher-order visual inputs do project to superficial layers of the lateral entorhinal cortex before reaching the hippocampus. These inputs modulate aspects of place cell expression, such as firing rates, remapping, or theta sequence dynamics such a preplay; but do not necessarily drive place cell formation. Our model focuses specifically on the visual encoding processes, which we argue are sufficient for generating place cells.
> > >
> > > The reviewer raises an important question: why doesn’t the visual cortex have place cells? Our analysis of representation dimensionality addresses this (Figure 3a): "We found that dense autoencoders had dimensionality scores close to 1, similar to visual cortex (Stringer et al., 2019), whereas sparse autoencoders exhibited higher-dimensional representations (Figure 3a), aligning more with the efficient coding hypothesis (Barlow et al., 1961)". Place cells necessitate high-dimensional representations (promoted by our orthonormal regularization term) that afford their localized spatial selectivity. This distinction explains why the visual cortex, with its dense lower-dimensional encoding, does not exhibit place cells, whereas our sparse autoencoders do.
> > >
> > > We hope these clarifications address the reviewer’s concerns and clarifies our contribution.

---

### Meta-Review · Area_Chair_83dB · 2024-12-19

**Metareview:**

This paper models the formation of hippocampal-like spatial representations by training sparse autoencoders on visual images from 3D environments. By applying an orthonormal activity regularization to encourage sparsity, the authors observe place cell-like behavior, where certain neurons respond selectively to specific regions in the input space. They argue that this sparsity enables the model to discretize continuous spaces into distinct memory-like representations. The study also extends to audio data, showing similar localized tuning, and evaluates how these representations impact reinforcement learning agents. The paper argues that such sparse coding mechanisms could illuminate how episodic memory and spatial representations emerge in the hippocampus.

The paper presents an interesting application of sparse autoencoders to simulate place cell-like behavior, providing valuable insights into how neural representations could emerge in the hippocampus. The approach is clearly written and utilizes a relevant, well-defined experimental setup that demonstrates promising results in both visual and auditory modalities.

The primary weakness lies in the confounding of visual and spatial representations. By using static visual images instead of a more dynamic, agent-driven approach to spatial navigation, the authors fail to accurately capture the integration of sensory and self-motion cues, weakening the connection to hippocampal place cell behavior. Additionally, the enforced sparsity via orthonormal regularization is not shown to improve reconstruction accuracy or task performance, raising concerns about the utility of these representations for spatial encoding. Furthermore, the paper lacks key experimental details, such as how the images were sampled, which hampers the understanding of how the model captures spatial information. Finally, the biological plausibility of the orthonormal regularization method and its comparison with other regularization techniques are not fully explored, limiting the paper’s impact.

Given these issues, the paper does not convincingly demonstrate how the proposed model advances our understanding of hippocampal encoding and memory formation, leading to the conclusion that it should be rejected.

**Additional Comments On Reviewer Discussion:**

During the rebuttal period, the authors responded to several key concerns raised by the reviewers, but several critical issues remained unresolved, ultimately leading to the rejection of the paper.

The authors attempted to address the ambiguity surrounding the use of the term "memory," particularly its relation to episodic memory. While they clarified their usage, the reviewer’s concerns about the term's narrow application and lack of a clear definition were not fully addressed. Similarly, the explanation of the model's zero-shot learning capability was considered superficial. While the authors discussed the clustering of images and rapid spatial emergence, the underlying mechanism was not sufficiently explained.

The authors also addressed concerns about the bio-plausibility and generalization of their model, particularly regarding hippocampal mechanisms like homeostatic plasticity. However, they failed to provide empirical evidence or simulations to support these claims, leaving the reviewer's concerns unresolved. For interpretability, the authors proposed adding visualizations, such as cosine similarity and SSIM scores, but this did not fully satisfy the concerns about making the model’s outputs clearer.

Additionally, the authors acknowledged limitations in their reinforcement learning experiments but did not conduct additional tests or provide sufficient evidence to address the concerns raised. Although the authors made efforts to clarify their unique contribution and distinguish their work from previous studies, they did not offer enough depth in this regard, leaving the novelty of the work partially unaddressed.

Given the unresolved concerns in these key areas, the paper was ultimately rejected.

---

### Decision · Program_Chairs · 2025-01-22

Reject